# Second-Hand Smoke Exposure and Risk of Lung Cancer Among Nonsmokers in the United States: A Systematic Review and Meta-Analysis

**DOI:** 10.3390/ijerph22040595

**Published:** 2025-04-10

**Authors:** Safa Elkefi, Gabriel Zeinoun, Achraf Tounsi, Jean-Marie Bruzzese, Corina Lelutiu-Weinberger, Alicia K. Matthews

**Affiliations:** 1School of Nursing, Columbia University, New York, NY 10027, USA; 2Automatic Data Processing, New Jersey, NJ 07932, USA

**Keywords:** second-hand smoke, lung cancer, cancer risk, tobacco, nonsmokers

## Abstract

This study aims to explore the link between exposure to tobacco smoke among nonsmokers and the risk of lung cancer in the United States. We searched six databases for studies on second-hand smoke (SHS) and lung cancer following PRISMA guidelines. Following the random effects model and specific statistical methods, our meta-analysis analyzed studies based on SHS exposure type. A total of 19 eligible studies were included in the review and 15 in the meta-analysis. We covered exposure from parents (childhood), spouses and partners (household), and work-related exposure (colleagues), with higher risk among non-smoking children and domestic partners. Findings reveal a consistent link between SHS exposure and increased lung cancer risk for this population (exposure effect sizes: 1.05–3.11). Analysis of childhood SHS exposure reveals a distinct increased risk associated with parental exposure. For nonsmokers living with smoking spouses, there is a marked 41% increase in risk. Higher risk was associated with more and more prolonged SHS exposure. Exposure to SHS in the workplace shows a correlation with lung cancer risk. Our findings highlight increased SHS-related lung cancer risk, particularly among non-smoking children and domestic partners, intensifying with the amount and duration of exposure, indicating the significant impact of SHS within domestic environments.

## 1. Introduction

Second-hand smoke (SHS) is a significant public health threat comparable to the risks of active smoking [1]. One out of three people in the world are passive smokers [1]. In the United States (US), between 2015 and 2018, approximately 20.8% of nonsmokers have been exposed to SHS [2], as measured by the presence of cotinine in their blood [2]. Notably, 25.6% of individuals between 18 and 39 had been exposed to SHS, compared to 19.1% of individuals between ages 40 and 59 and 17.6% aged 60 and older [2]. SHS exposure was more prevalent among Black adults (39.7%) compared with Hispanic (17.2%) and White (18.4%) adults [2]. Conclusive scientific evidence documents that SHS is harmful to nonsmokers [3,4,5]. Numerous studies conducted across different continents supported the causality between SHS and lung cancer [6]. Furthermore, in adults, SHS has direct adverse cardiovascular issues, while long-term exposure can lead to the development of heart diseases, strokes, and sometimes death [5]. Exposure to SHS puts children at higher risk of sudden death syndrome, respiratory infections, worsening asthma, respiratory symptoms, and slowed lung growth [5].

Many research studies have investigated the link between SHS and being exposed to smoke at home and in the workplace. While smoke-free policies usually focus on public and work settings, fewer countries have implemented smoking exposure regulations in private or semi-private spaces, such as public housing and vehicles [7]. Considering the high likelihood of males smoking (fathers/spouses), children and women may benefit less from smoke-free measures in non-public spaces [7,8]. Many individuals encounter difficulties establishing rules prohibiting smoking within the household or reducing exposure to SHS emanating from adjacent residences [9] and often resort to measures (e.g., smoking only in some regions of the residence or smoking through open windows) that remain less effective [9]. Non-smoking households in multiunit housing face these particular challenges, as public smoking bans may not fully protect them from SHS [10]. Future research should also investigate the long-term effects of workplace (and other public place smoking bans, like bars and restaurants) smoking bans on lung cancer risk.

Prior studies have confirmed the causal link between SHS exposure and lung cancer [6,11,12]. Fewer have examined the contexts in which SHS exposure occurs and how these may impact risk differently. Understanding where individuals are most exposed—at home, work, or during childhood—is essential for designing targeted prevention strategies. Several meta-analyses and systematic reviews have evaluated the general association between SHS and lung cancer [6,11,12]. Nevertheless, the aforementioned studies lacked differentiation between exposure settings, thereby complicating the identification of specific locations that significantly contribute to lung cancer risk. This review enhances prior research through the following methodologies: (1) stratifying second-hand smoke (SHS) exposure by specific settings (such as childhood, household, and workplace) to ascertain which environments present the highest risk; (2) conducting a meta-analysis that isolates various types of exposure (such as childhood, household, and workplace) permitting a more accurate evaluation of risk variation across different contexts; and (3) addressing the heterogeneity prevalent in prior findings by accounting for varying exposure durations and intensities. By addressing this research gap, the current study offers pragmatic insights for policymakers and healthcare practitioners in prioritizing their public health initiatives aimed at mitigating SHS exposure.

Furthermore, this study is confined to the United States for several reasons. One is that smoking behaviors and patterns of SHS exposure exhibit considerable variation on a global scale, complicating the validity and feasibility of international comparisons. Additionally, the smoking regulatory frameworks, including smoke-free public spaces (e.g., workplace, public transportation) legislation and housing regulations, diverge substantially across nations. Focusing exclusively on US data guarantees a locally relevant policy context, which is not universally generalizable. Ultimately, public health recommendations necessitate a country-specific approach, as findings derived from a U.S.-centric perspective enable the formulation of highly contextualized policies and intervention strategies devoid of the complexities introduced by global regulatory disparities.

## 2. Materials and Methods

A literature review is conducted, followed by a meta-analysis based on a selection of these articles. Our systematic review of the literature followed the PRISMA guidelines (Preferred Reporting Items for Systematic Reviews and Meta-Analysis) [13,14].

### 2.1. Search Strategy

We searched through six databases to gather pertinent publications published up to December 2023: Google Scholar, PsycINFO, Scopus, PubMed, Web of Science, and IEEE Xplore. These databases were selected based on their coverage of key research areas related to SHS exposure, lung cancer risk, and environmental health impacts [15]. For instance, PubMed provides biomedical and epidemiological research, ensuring coverage of clinical and public health studies on SHS and lung cancer. Scopus and Web of Science index high-impact journals across public health, epidemiology, occupational health, and environmental sciences, ensuring comprehensive coverage of peer-reviewed literature. PsycINFO captures research on behavioral aspects of SHS exposure, including risk perception and psychosocial impacts. IEEE Xplore includes air quality monitoring, environmental health, and occupational SHS exposure studies. Finally, Google Scholar captures studies that may not be indexed in traditional databases, ensuring search comprehensiveness. By selecting these databases, we ensured broad disciplinary coverage and maximized the inclusion of relevant studies. The search incorporated keywords relevant to SHS and lung cancer (Appendix A, Figure A1). Experts and librarians reviewed the keywords.

### 2.2. Inclusion Criteria

We included studies that (1) were empirical (quantitative or qualitative), (2) reported outcomes specifically related to lung cancer, (3) examined the association between SHS exposure and lung cancer risk, (4) were peer-reviewed, (5) were written in English, and (6) were conducted in the US. Studies focusing solely on other malignancies caused by tobacco exposure, such as head and neck cancers, were excluded given the focus of the present inquiry.

### 2.3. Study Selection

Among the selected studies, only those that reported quantitative findings on the association between SHS exposure and lung cancer risk were included in the quantitative analysis (meta-analysis). The selection was conducted in three stages, following PRISMA guidelines. First, following removal of duplicates, three independent reviewers screened titles and abstracts for relevance. Next, the remaining articles underwent a full-text review by the same three reviewers to assess eligibility. Finally, studies that met all inclusion criteria were selected for the systematic review and, where applicable, for the meta-analysis. Discrepancies at each step were discussed until a consensus was reached. If disagreements persisted, a fourth senior reviewer was consulted to make the final decision. This approach ensured that selection decisions were robust, unbiased, and consistent.

### 2.4. Data Extraction and Quality Assessment

The team collaboratively developed a data extraction form to capture the data for our research questions, which was pilot tested on five articles and refined iteratively (see Appendix B, Table A1).

Following the meta-analysis practices [16], we separated the study analysis based on the area of exposure reported (household, childhood, work, all types together) to prevent multiple weighting of the same study. We used the Joanna Briggs Institute’s (JBI) critical appraisal checklists to evaluate the risk of bias in each study [17].

### 2.5. Data Analysis

Following Masoumi et al.’s study [18], we conducted a meta-analysis to estimate the pooled effect size of second-hand smoke (SHS) exposure on lung cancer risk, using log-transformed odds ratios (ORs) with 95% confidence intervals (CIs). Given the expected heterogeneity among studies (differences in populations, exposure assessment methods, and study designs), we used a random-effects model, which assumes that individual studies estimate different but related actual effects rather than a single common effect [17]. This model was chosen over a fixed-effects model because it accounts for both within-study and between-study variability, making it more suitable when study characteristics vary significantly. The random-effects model ensures that our findings are more generalizable across different populations and study conditions, allowing for variability in the underlying effect size.

We applied the Hartung–Knapp–Sidik–Jonkman (HKSJ) adjustment to improve confidence interval accuracy, which is particularly useful in meta-analyses with moderate-to-high heterogeneity and a small-to-moderate number of studies. Compared to conventional random-effects models, HKSJ (1) adjusts for uncertainty in variance estimation, reducing the risk of overly narrow confidence intervals, which can lead to overconfident conclusions; (2) reduces type I error rates, making it more reliable in scenarios where standard random-effects models might overestimate precision; (3) and provides more accurate estimates when the number of included studies is relatively small, often in exposure-based epidemiological meta-analyses.

We assessed statistical heterogeneity using Cochran’s Q and I^2^ statistics, with I^2^ values classified as low (≤25%), moderate (26–50%), and high (>50%) heterogeneity [18,19]. High heterogeneity suggests substantial variability in effect sizes across studies, which justifies using a random-effects model. Publication bias was evaluated using funnel plots and Egger’s test [19,20,21,22] to determine whether small-study effects influenced our findings. To assess the robustness of our results, we performed sensitivity analyses using the leave-one-out method, systematically excluding each study one at a time to evaluate its influence on the overall effect estimate. Additionally, we explored potential sources of heterogeneity through subgroup analyses based on exposure type (childhood, household, workplace) and study design. More details are in Appendix A.

## 3. Results

### 3.1. Study Identification

We first identified 54,429 articles (Figure 1). After screening, 19 studies were eligible for our review (Table A2, Appendix C), and 15 were eligible for the meta-analysis (Appendix E).

### 3.2. Study and Participant Characteristics for the Systematic Review

Appendix B (Table A1) summarizes the included studies’ baseline characteristics, published between 1983 and 2023. Study designs varied: nine were case–control, five were cohort, two were cross-sectional surveys, two were longitudinal surveys, and one was qualitative.

Four studies had national samples, while others were conducted in different US regions: Northeast (*n* = 5); Midwest (*n* = 5); South (*n* = 6); and West (*n* = 4). Most studies (18/19) adjusted for gender, and seven included only women.

Sample sizes varied, with case–control studies ranging from 134 to 1338 participants, cohort studies from 810 to 133,385, cross-sectional surveys from 47 to 49,569 participants, and longitudinal studies from 19,286 to 41,632 nonsmokers. An interview-based study had 537 participants.

Participant demographics also varied. Six studies did not adjust for race, two did not account for age, and three focused only on women, adjusting for age. Three studies reported findings for specific populations (White women, Black women), all adjusting for age. Six studies did not report participants’ socioeconomic status.

### 3.3. Quality Assessment for the Systematic Review

Following the quality assessment results, we deemed all studies sufficiently adequate in quality to be included in the systematic review (Appendix D). All studies surpassed the quality threshold (i.e., eight in JBI).

### 3.4. Systematic Review Findings on the Risk of Lung Cancer per Exposure Area to SHS

#### 3.4.1. Lung Cancer Risk Associated with Overall Exposure to SHS (All Sources Together)

Eight studies reported findings related to overall SHS exposure (see Appendix Cand Appendix E). Three studies found a positive association with lung cancer risk [23,24,25], with effect sizes ranging from 1.05 to 3.11. Only one study found a non-significant relationship [26]. Only one of these studies focused on Black populations’ exposure to SHS and was limited to Black women. One study showed that the risk of exposure is higher starting from five years of continuous exposure (OR = 1.28) [24], while another study found that lifetime exposure to SHS was associated with increased lung cancer risk (OR = 1.14) [25]. Another study investigating long-term exposure showed that the more non-smoking people are exposed to tobacco smoke, the higher the risk of lung cancer (Exposure <20 years: OR = 1.09; Exposure 20–38 years: OR = 1.21; Exposure >38 years: OR = 1.32) [27]. For multiple types of SHS exposure, a combined exposure of a minimum of 10 years during childhood, 30 years during adulthood, and 20 years of work exposure resulted in the highest risk in (OR = 1.76) [28]. It is noteworthy that the study by Bastian et al. showed that women veterans had higher rates of exposure to passive smoke but did not show a higher adjusted risk for lung cancer compared to non-veterans [29].

#### 3.4.2. Lung Cancer Risk Associated with Childhood Exposure to SHS

Nine studies reported findings related to childhood exposure to SHS. We included six of these in the meta-analysis.

##### Both Parents Smoking

Seven of the studies reported findings related to both parents smoking (see Appendix C and Appendix E), but only four were subsequently included in the meta-analysis. Two studies investigated both parents’ history of smoking, both of which showed a positive correlation between SHS exposure and lung cancer risk [25,30]. Additionally, a study by Janerich et al. found that people who had prolonged exposure of up to 21 years of age had twice the risk of lung cancer [25]. The findings by Tyc et al. showed that 58% of the parents whose children had lung cancer smoked inside the home, with higher exposure among older children [31]. Another study revealed a higher risk of lung cancer with a higher intensity of exposure, although the associations were not significant [26]. Similarly, no clear associations were found in the study by Bastian et al. [29].

##### Mothers or Fathers Smoking

Three studies assessed the difference in children’s SHS exposure from mothers and fathers [23,30,32]. They each showed that children whose mothers smoked were at higher risk of lung cancer (OR = 1.60–2.92) compared to the risk associated with fathers smoking (OR = 1.04–2.89) [23,30,32].

#### 3.4.3. Lung Cancer Risk Associated with Household Exposure to SHS

Fourteen studies examined the risks related to household exposure, particularly by spouses and partners. We included 11 in the meta-analysis, which indicated that significant lung cancer-related risks were associated with passive smoking among married couples [24,33] and showed that women who were married to husbands who smoked more than forty cigarettes per day or who were exposed to the smoke of at least 20 cigarettes per day at home showed a doubled risk compared to non-exposed women. A study comparing household exposure to childhood SHS exposure showed that high levels of environmental tobacco smoke exposure in adulthood among non-smoking women increased their lung cancer risk by approximately 30% [26]. In contrast, the same study noted no consistent elevated risk among the same population for childhood or workplace exposure [26]. Rates of lung cancer deaths were 20% higher among spouses with smoking husbands compared to never-smokers, interestingly with a higher relative risk among women whose husbands smoked compared to when the wives were smoking (RR 1.6 vs. 1.1) [34].

#### 3.4.4. Lung Cancer Risk Associated with Workplace Exposure to SHS

Seven studies covered work-related SHS exposure, of which four were included in the meta-analysis. All the included studies showed a positive association with lung cancer risk, with a higher risk associated with more prolonged exposure [27,28,35]. For instance, a study by Ref. [27] reported that the risk of lung cancer increased from OR = 1.04 to OR = 1.20 and OR = 1.26 for less than eight years, 8 to 20 years, and above 20 years of exposure, respectively [27]. Ref. [35] demonstrated significant health risks from exposure to SHS in restaurants and bars (OR = 1.22). Rates of lung cancer death for servers and patrons were well above acceptable levels due to SHS exposure in these settings [35]. Another study investigating occupational exposure among Black women showed that SHS significantly increased lung cancer risk (sHR = 1.93, *p* = 0.006) [36].

### 3.5. Meta-Analysis Results

Below, we report findings from the meta-analysis conducted per exposure area (household, childhood, and workplace-related exposures) for the 15 included studies. Four studies were excluded for different reasons; one reported findings related to the occurrence of secondary lung cancer, which can represent a bias to our findings on the risk of primary lung cancer [37]. Studies focusing on secondary (metastatic) lung cancer were excluded because their outcomes do not represent the risk of developing lung cancer due to SHS exposure but rather the spread of cancer from other primary sites. Other studies reported their outcomes by comparing the risk among passive smokers compared to active smokers instead of non-smoking people who are exposed vs. not exposed to SHS [29,31,33]. While these studies provide insights into lung cancer risk, the way the data are presented does not allow use for meta-analysis purposes [29,31,33].

#### 3.5.1. Lung Cancer Risk Associated with Overall Exposure to SHS

The combined effect size in this analysis is 1.08. The proximity of the confidence interval to 1 indicates that, while there may be an effect, it might be of a lower magnitude than what could be considered impactful or definitive. Moreover, there is considerable heterogeneity among the studies, as indicated by an I^2^ of 67.08%, suggesting that while there is a general trend towards a positive effect, the studies vary significantly in the magnitude of that effect. Cochran’s Q statistic of 21.26 with 7 degrees of freedom reinforces this observation of notable heterogeneity among the studies, possibly due to various factors such as different study designs.

#### 3.5.2. Lung Cancer Risk Associated with Childhood Exposure to SHS

The combined effect size of exposure by both parents was OR = 1.40, suggesting a statistically significant association between the exposure or condition studied and the lung cancer risk. Despite this significant combined effect (*p* < 0.001), there is substantial heterogeneity among the studies (I^2^ = 84.77%), suggesting that while the studies are pointing in the same general direction (i.e., an effect exists), they differ significantly in the magnitude of that effect. Cochran’s Q statistic of 19.69 suggests low heterogeneity among the studies, which suggests that the studies are somewhat consistent. The sensitivity analysis suggests that all scenarios show high heterogeneity, indicating that the studies are quite different, regardless of which one is excluded. Although results change when excluding the different studies, the overall combined effect size always indicates a link between the nonsmokers’ risk of lung cancer and SHS exposure.

For mothers’ exposure, while there appears to be a considerable positive association between exposure and risk when the studies are combined, this effect is not statistically significant across the three studies included in this meta-analysis (OR = 1.40; *p* = 0.24). The studies have moderate heterogeneity (I^2^ = 28.48%), indicating that the study results are not drastically different. The sensitivity analysis suggested that excluding different studies changes the combined effect size and levels of heterogeneity significantly. The study by Olivio-Martson et al. [30] seems fundamentally different from the other two, as excluding it results in no observed heterogeneity among the remaining studies and lowers the combined effect size. That study is the only one suggesting a significant positive association between the risk of lung cancer and SHS exposure to fathers’ smoking.

For exposure to fathers’ smoking, the combined effect size from the three studies is 1.22 and the pooled effect is significant (*p* = 0.028). However, the studies have high heterogeneity (I^2^ = 72.04%), meaning the study results vary substantially. The Cochran’s Q of 7.15 with 2 degrees of freedom also suggests variability among the study outcomes. The sensitivity analysis shows that all combined effect sizes are more significant than 1, which generally supports the hypothesis that exposure to fathers’ smoking is linked to a higher risk of lung cancer among nonsmokers. Given the high effect sizes and significant findings, there seems to be a robust association between exposure to fathers’ smoking and increased lung cancer risk across different contexts. However, the significant heterogeneity when excluding different studies suggests that the exact magnitude of this risk may vary based on specific conditions or populations.

#### 3.5.3. Lung Cancer Risk Associated with Household Exposure to SHS

When looking at the combined effect of all the studies that reported household SHS exposure, we found that overall household exposure is associated with 41% more risk when compared to unexposed individuals (*p* < 0.001). The high Q value of 51.57 indicates substantial heterogeneity among the results of the different studies. Despite the significant association, the high heterogeneity (I^2^ = 80.61%) suggests that the study outcomes differ substantially. This variation could be due to differences in study populations, methods, exposures, or other factors. In sensitivity analyses, where each study was excluded to examine its influence on the overall effect estimate, we observed consistent results that support the association between SHS exposure and an increased risk of lung cancer. The combined estimates ranged from 1.39 to 1.52 when each study was excluded, indicating that no single study disproportionately influenced the overall effect estimate. For instance, when excluding the study by Correa et al. [23], the combined effect size was OR = 1.39, indicating a significant association between SHS exposure and lung cancer. Similarly, excluding the study by Ref. [24] yielded a combined effect size of 1.48. This pattern was consistent across all excluded studies, underscoring the robustness of our findings. The I^2^ values ranged from 69.58% to 82.87%, suggesting substantial heterogeneity. It is important to note that Cochran’s Q statistic was significant in all cases, confirming the presence of heterogeneity (see Figure 2).

#### 3.5.4. Lung Cancer Risk Associated with Work Exposure to SHS

The combined effect size for the association between SHS exposure in workplaces and lung cancer risk among nonsmokers was OR = 1.24, indicating an increased risk of lung cancer associated with SHS exposure. Moderate heterogeneity among the studies (I^2^ = 41.05%) does not undermine the overall conclusion (since the *p*-value for Cochran’s Q suggests this variability could be due to chance). The moderate heterogeneity also indicates that the strength of this effect might vary slightly among different studies or populations. An I^2^ of 41.05% represents moderate heterogeneity.

## 4. Discussion

In this study, we investigated the impact of SHS exposure on lung cancer risk among nonsmokers. Our findings suggest a demonstrated association, with more prolonged exposure (>5 years) resulting in higher risk. These results align with the literature on the risks of SHS [6,41].

### 4.1. Childhood Exposure to SHS as a Risk of Lung Cancer

Our study showed that a higher risk of lung cancer was associated with both parents smoking [25,30]. This finding is consistent with previous studies on how parental smoking could harm children’s respiratory health [42,43]. Even when only one of the parents was smoking, the risk persisted [23,30,32]. However, mothers’ smoking resulted in a higher risk of lung cancer [23,30,32]. The time children spend with their mothers tends to be longer than that spent with fathers, which may explain this result [44,45]. Research has shown that time spent with a smoking parent significantly correlated with worse health outcomes in children [46].

Despite these findings, our sensitivity analyses revealed high heterogeneity among studies examining childhood SHS exposure. This heterogeneity persisted even after excluding individual studies, suggesting that the observed differences cannot be attributed to a single outlier. Several factors may explain this variability, such as differences in exposure measurement. For instance, some studies quantified SHS exposure using parental smoking status alone, while others considered intensity (cigarettes/day or pack-years) and duration (years of exposure). The observed heterogeneity across studies could also be related to the variability in study designs. For instance, case–control and cohort studies may yield different risk estimates due to differences in how participants are selected and recall bias in retrospective assessments. Also, it could be related to the recall bias in childhood exposure reporting. For instance, since lung cancer has a long latency period, exposure recall may be less precise, particularly in studies relying on self-reported childhood exposure from adults.

### 4.2. Household Exposure to SHS as a Risk of Lung Cancer

Our study revealed that women who do not smoke but are considerably exposed to tobacco smoke from their spouses (i.e., several cigarette packs/year and time) are at more risk of developing lung cancer compared to women not exposed at all [24], which is consistent with previous epidemiological studies [47]. Previous research has suggested that men have a higher risk of developing lung cancer [48]. Although previous studies do not support the higher female susceptibility to tobacco-related lung cancer, it is noteworthy that a study by Cardenas et al. found that exposure to spouse risk results in a higher chance of lung cancer among women compared to men [34]. This result suggests that while women might not be inherently more susceptible to tobacco-related lung cancer from personal smoking, they could be more vulnerable to the effects of SHS from their spouses compared to men. Further investigation is needed to validate this result.

Additionally, some studies showed that household exposure was associated with worse lung cancer outcomes than childhood and workplace exposures [24,26]. The authors elucidated two potential explanations. Firstly, the passive smoke-related risk of lung cancer during childhood may decline by adulthood, especially in the absence of adulthood exposure [26]. Secondly, there is potentially low reliability in the quantitative measures (both intensity and duration) of passive smoke exposure during childhood, rendering the assessment of lung cancer risk attributable to childhood passive smoke exposure incredibly challenging [26,49,50,51]. The same issue was attributable to the measures used to self-report exposure in the case of workplace exposure [52].

### 4.3. Workplace Exposure to SHS as a Risk of Lung Cancer

Our findings indicated a positive association between workplace-related SHS exposure and lung cancer risk, which strengthens with prolonged exposure [27,28,35]. It is noteworthy that although the association is positive, only the study by [35] investigated exposure to a specific environment of work (restaurants and bars), and one study by [36] reported associations for a specific population (Black women). All other studies reported general occupational exposure findings among all participants [27,28,36]. Additionally, no studies compared the exposure risk across different types of jobs. Research has differentiated the SHS exposure levels based on job type: indoor vs. outdoor, number of employees in the workplace, number of smoking employees, and field of occupation (mining, tourism, manufacturing, etc.) [53]. Considering that the difference in jobs may result in different risk outcomes, future research should incorporate this classification into workplace exposure assessment.

### 4.4. Practical Implications

Our results emphasize the importance of implementing efforts that minimize nonsmokers’ exposure to SHS at home and in the workplace. Regarding reducing risk for home SHS, clinicians could incorporate standardized questions about SHS exposure, particularly household exposure, into routine patient history intake, especially for individuals with no personal history of smoking, including children and adolescents. Relatedly, it is essential to consider expanding lung cancer screening eligibility criteria to include non-smoking individuals with significant SHS exposure, particularly those with additional risk factors. Prevention of SHS and potential subsequent lung cancer remains a critical public health priority and, based on findings from this study, family smoking cessation interventions may also be appropriate to raise awareness among household members and promote harm reduction strategies.

Public health efforts should focus on educating non-smoking women about the negative impacts of household SHS exposure, emphasizing the importance of establishing smoke-free zones within the home. One of our studies indicated that SHS exposure during childhood, particularly from both parents, increases lung cancer risk. However, the risk from childhood exposure may decline by adulthood if exposure does not continue, underscoring the need to protect children from SHS to reduce long-term cancer risk.

Workplace SHS exposure also presents a significant risk, particularly in service industries such as restaurants, bars, and casinos, where smoking remains relatively more prevalent. Policies are needed and/or enforced through strict accountability specifications for employers to implement stricter smoke-free workplace policies, create designated outdoor smoking areas, and ensure adequate ventilation when exposure is difficult to prevent. Additionally, further research is needed to assess the variation in SHS exposure across different job types, neighborhoods, and workplace environments, as factors such as the number of smoking colleagues and shared workspace design that may influence risk levels.

Societal responsibility in enforcing smoke-free policies should be prioritized at a broader level. Multiunit housing regulations should be strengthened and enforced to prevent SHS infiltration between residences through shared ventilation systems and common spaces. Expanding smoke-free policies to include outdoor public spaces like parks, bus stops, beaches, and restaurant patios could further protect nonsmokers. Public awareness campaigns should be implemented at the community level to educate individuals about the dangers of SHS exposure and the importance of complying with smoking restrictions. By integrating individual responsibility with policy enforcement, a comprehensive approach to SHS reduction can be achieved, ultimately reducing lung cancer risk in nonsmokers.

### 4.5. Limitations

The are some limitations to this review that we should acknowledge. Considering the differences in laws related to smoking in different countries, we excluded studies focusing on countries other than the US, which limits the generalizability of our findings. Additionally, we excluded articles written in languages other than English, which may have resulted in missing some studies. Furthermore, several included studies examined socioeconomic and racial/ethnic disparities in SHS exposure and lung cancer risk. However, not all studies accounted for SES and racial factors, similarly limiting our ability to conduct a detailed pooled comparative analysis across these demographic variables. Some studies adjusted for income, education, and housing status, while others only stratified by broad racial groups without accounting for socioeconomic factors. A few studies examined occupational class and workplace exposure disparities, but methods varied significantly, making direct comparisons difficult. Several studies did not adjust for SES at all, meaning potential confounding effects remain unaccounted for in some findings. While some studies used the number of packs/years to evaluate the high exposure, others considered the number of years exposed to smokers. Effectively measuring SHS is crucial to offering precise, timely, and cost-efficient ways of evaluating exposure intensity. In addition, many included studies relied on self-reported SHS exposure, subject to recall bias and misclassification errors. Individuals may underestimate or overestimate their exposure based on memory, leading to potential measurement inaccuracies. Studies using biomarker-based SHS assessment (e.g., cotinine levels) generally reported stronger associations between SHS and lung cancer risk, suggesting that self-reported data may dilute actual risk estimates. Also, the studies included in this review may be biased toward publishing significant findings, while studies with null or weak associations may be underreported. Certain studies had non-representative samples (e.g., hospital-based case–control studies), which may not reflect broader population-level effects. Finally, while the overall meta-analysis showed a positive association between SHS exposure and lung cancer risk, some studies reported weaker or non-significant associations. Possible explanations may include exposure misclassification, not adequately adjusting for active smoking, and variability in the exposure assessment methods.

## 5. Conclusions

This study reinforces the strong association between SHS exposure and lung cancer risk, emphasizing the need for comprehensive interventions to limit exposure in households, workplaces, and public spaces. The findings support stricter SHS policies, mainly through smoke-free housing regulations, workplace protections, and the expansion of outdoor smoke-free areas.

Despite strong evidence linking SHS to lung cancer, several critical gaps remain that warrant further investigation. Future studies should explore longitudinal trends in SHS exposure to determine how cumulative and long-term exposure influences lung cancer risk. Prospective cohort studies integrating biomarker-based assessments (e.g., cotinine levels) would improve exposure measurement accuracy and strengthen causal inferences. Additionally, emerging research suggests that genetic predisposition may influence individual susceptibility to SHS-related lung cancer. Investigating gene–environment interactions could help identify high-risk populations and inform personalized prevention strategies.

Moreover, more research is needed to examine racial, ethnic, and socioeconomic disparities in SHS exposure and lung cancer risk. While some studies adjust for these factors, many do not consistently account for these important social determinants of health as predictors for SHS exposure and lung cancer, limiting the ability to conduct in-depth analyses. Future studies should incorporate biomarker-based SHS assessments to reduce reliance on self-reported exposure, which is prone to recall bias. Additionally, studies should compare SHS exposure risk across different workplace environments, focusing on factors such as industry type, ventilation conditions, and workforce smoking prevalence. By addressing these policies and research gaps, public health strategies can be further refined to prevent SHS-related lung cancer and enhance protective measures for at-risk populations.

## Figures and Tables

**Figure 1 ijerph-22-00595-f001:**
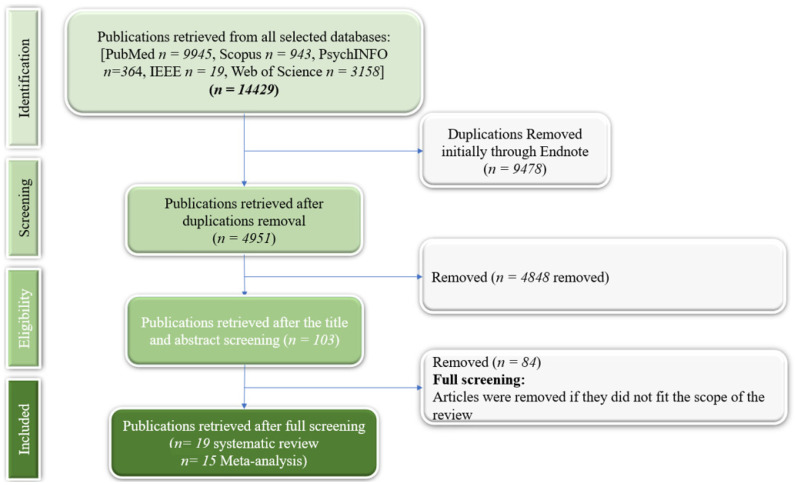
PRISMA flowchart.

**Figure 2 ijerph-22-00595-f002:**
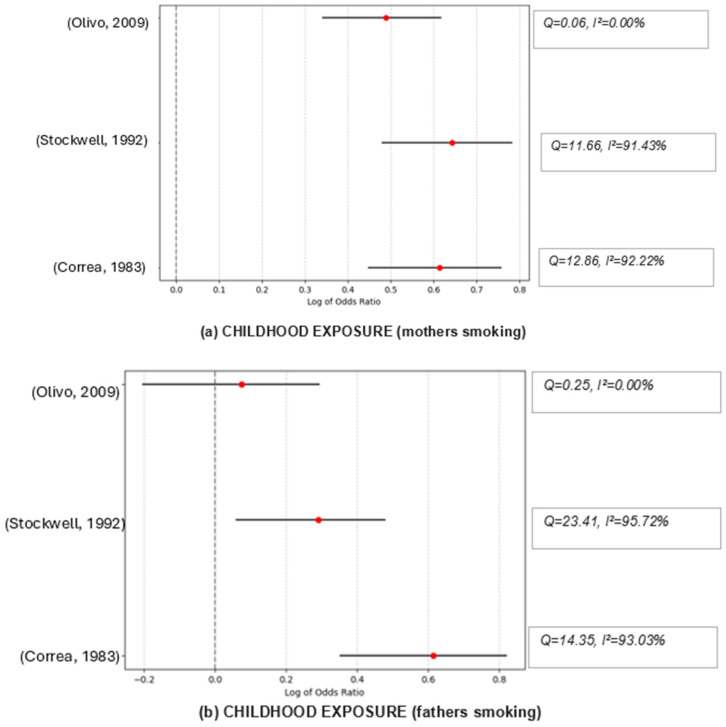
Funnel graphs for the excluded studies one at a time (sensitivity analysis). Childhood exposure (mothers smoking): Olivo, 2009 → [30], Stockwell, 1992 → [32], Correa, 1983 → [23]; Childhood exposure (fathers smoking): Olivo, 2009 → [30], Stockwell, 1992 → [32], Correa, 1983 → [23]; Childhood exposure (both smoking): Abdel-Rahman, 2020 → [38], Olivo-Marston, 2009 → [30], Brownson, 1992 → [26], Janerich, 1990 → [25]; Work exposure: Erhunmwunsee, 2022 → [36], Abdel-Rahman, 2020 → [38], Liu, 2014 → [35], Brennan, 2004 → [27]. Household exposure: Erhunmwunsee, 2022 → [36], Abdel-Rahman, 2020 → [38], Asomaning, 2008 → [39], Brennan, 2004 → [27], Cardenas, 1997 → [34], Alavanja, 1995 → [40], Wang, 1995 → [28], Stockwell, 1992 → [32], Janerich, 1990 → [25], Garfinkel, 1985 → [24], Correa, 1983 → [23].

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
