# Peer review of "Second-Hand Smoke Exposure and Risk of Lung Cancer Among Nonsmokers in the United States: A Systematic Review and Meta-Analysis"

_ijerph, 2025, doi:10.3390/ijerph22040595_

Round 1
Reviewer 1 Report
Comments and Suggestions for Authors
Safa et al., investigate the association between secondhand smoke exposure and lung cancer risk among U.S. nonsmokers. The authors systematically reviewed and compiled nearly 55,000 literature, and used rigorously statistical methods to analyze the association between second-hand smoke exposure in different exposure pathways (childhood, family, workplace) and the risk of lung cancer. The results showed that exposure to second-hand smoke, particularly in childhood and family environments, will significantly increase the risk of lung cancer in non-smokers, and the longer the exposure time correlates the higher the risk. The findings highlight the need for greater public health education and policies to reduce non-smokers' exposure to secondhand smoke. The contribution of this paper included:
- While past research has examined the link between secondhand smoke and lung cancer risk, this study specifically focused on distinguishing the effects of different exposure areas, such as home, childhood, and the workplace.
- Deeper analysis of exposure types: The study differentiated between second-hand smoke exposure from parents in childhood and further differentiated the effects of exposure from mothers and fathers, finding that maternal smoking had a greater impact on children's lung cancer risk.
- Research highlights the risks of prolonged exposure to secondhand smoke, noting that exposure in specific work environments such as restaurants and bars is particularly harmful. For example, household secondhand smoke exposure increases lung cancer risk by 41%. Non-smokers who live with a smoking spouse have a 20 percent higher lung cancer death rate.
- This study further clarifies the impact of different types of secondhand smoke exposure on different genders. For example, women are at a higher risk of being exposed to secondhand smoke from their spouses. The research looks specifically at the effects of exposure during childhood and the potential harm to children from maternal smoking.
Generally, this article is well prepared, the guidelines are clear, the static methods are rigorous, and also connect epidemiological investigation and clinical improvement. The evidence is solid and clear.
Some concern issues were raised:
- Only studies conducted in the United States were included in this study. Because smoking-related laws and policies vary between countries, the findings may not be directly generalizable to other countries. A particular country or region might be indicated in the title or abstract.
- Although studies performed subgroup analyzes for different exposure types, unexplained heterogeneity remained. For example, findings from studies of childhood exposures show a high degree of heterogeneity. Even in sensitivity analyses that exclude individual studies, heterogeneity issues persist, suggesting that between-study differences are not due to a single study. The authors might need to interpret this in the discussion section.
- Is It possible to analyze the risk of secondhand smoke exposure posed by different types of work? For example, different workplaces (e.g., indoor vs. outdoor, number of employees, number of employees who smoke), may result in varying levels of exposure, and these differences have not been fully explored.
Author Response
REVIEWER 1
Safa et al., investigate the association between secondhand smoke exposure and lung cancer risk among U.S. nonsmokers. The authors systematically reviewed and compiled nearly 55,000 literature, and used rigorously statistical methods to analyze the association between second-hand smoke exposure in different exposure pathways (childhood, family, workplace) and the risk of lung cancer. The results showed that exposure to second-hand smoke, particularly in childhood and family environments, will significantly increase the risk of lung cancer in non-smokers, and the longer the exposure time correlates the higher the risk. The findings highlight the need for greater public health education and policies to reduce non-smokers' exposure to secondhand smoke. The contribution of this paper included:
- While past research has examined the link between secondhand smoke and lung cancer risk, this study specifically focused on distinguishing the effects of different exposure areas, such as home, childhood, and the workplace.
- Deeper analysis of exposure types: The study differentiated between second-hand smoke exposure from parents in childhood and further differentiated the effects of exposure from mothers and fathers, finding that maternal smoking had a greater impact on children's lung cancer risk.
- Research highlights the risks of prolonged exposure to secondhand smoke, noting that exposure in specific work environments such as restaurants and bars is particularly harmful. For example, household secondhand smoke exposure increases lung cancer risk by 41%. Non-smokers who live with a smoking spouse have a 20 percent higher lung cancer death rate.
- This study further clarifies the impact of different types of secondhand smoke exposure on different genders. For example, women are at a higher risk of being exposed to secondhand smoke from their spouses. The research looks specifically at the effects of exposure during childhood and the potential harm to children from maternal smoking.
Generally, this article is well prepared, the guidelines are clear, the static methods are rigorous, and also connect epidemiological investigation and clinical improvement. The evidence is solid and clear.
We thank you for your comments and feedback.
Some concern issues were raised:
- Only studies conducted in the United States were included in this study. Because smoking-related laws and policies vary between countries, the findings may not be directly generalizable to other countries. A particular country or region might be indicated in the title or abstract.
We thank you for your comment and clarified that in the title and abstract.
- Although studies performed subgroup analyzes for different exposure types, unexplained heterogeneity remained. For example, findings from studies of childhood exposures show a high degree of heterogeneity. Even in sensitivity analyses that exclude individual studies, heterogeneity issues persist, suggesting that between-study differences are not due to a single study. The authors might need to interpret this in the discussion section.
We thank you for your insightful comment regarding the unexplained heterogeneity in childhood exposure studies. We acknowledge that, despite subgroup analyses, substantial between-study heterogeneity remains, particularly in childhood SHS exposure.
To address this, we have revised the Discussion section to:
- Acknowledge the persistent heterogeneity in childhood exposure studies.
- Explain potential sources of heterogeneity, such as:
Variability in study designs, populations, and exposure measurement methods.
Differences in how childhood SHS exposure was recalled and reported across studies.
The impact of additional environmental and social factors not consistently accounted for in the studies.
Propose future research directions to reduce heterogeneity, such as standardizing exposure measurement and conducting meta-regression analyses.
- Is It possible to analyze the risk of secondhand smoke exposure posed by different types of work? For example, different workplaces (e.g., indoor vs. outdoor, number of employees, number of employees who smoke), may result in varying levels of exposure, and these differences have not been fully explored.
Thank you for your insightful comment. We acknowledge that workplace exposure to secondhand smoke (SHS) is not uniform across different job settings and that factors such as indoor vs. outdoor environments, workplace size, and the number of smoking employees may significantly influence exposure levels. Unfortunately, the studies included in our review did not provide granular data on workplace characteristics beyond broad occupational exposure categories. Most studies assessed overall workplace exposure rather than stratifying risk based on job type, workplace ventilation, or the density of smokers in the environment.
Reviewer 2 Report
Comments and Suggestions for Authors
The study investigates the relationship between second-hand smoke (SHS) exposure and lung cancer risk among non-smokers in the United States through a systematic review and meta-analysis. The topic is highly relevant for public health and tobacco control policies. The authors follow PRISMA guidelines and conduct a meta-analysis using robust statistical methods. However, the paper has several areas that require improvement before it can be considered for publication.
The introduction presents a clear rationale for the study, emphasizing the health risks associated with SHS. The authors provide statistics on SHS exposure in the U.S. and outline its impact on different demographic groups. However, the introduction could be improved by:
- Clearly defining the research gap (e.g., how this study builds upon previous meta-analyses).
- Expanding on the novelty of the study in comparison to prior systematic reviews.
- Providing a stronger justification for focusing solely on the U.S. rather than including global data.
The methodology is thorough and follows PRISMA guidelines, which is a strength. However, there are several weaknesses:
- The authors searched five databases, but there is no explanation of why these specific databases were chosen over others.
- The selection process for studies (including resolving reviewer disagreements) could be elaborated further.
- The use of statistical methods (random effects model, Hartung-Knapp method) is appropriate, but the rationale for choosing these specific methods should be better justified.
The results section is well-structured and provides a clear summary of study identification, participant characteristics, and quality assessment. However, the following improvements are needed:
- There is a significant level of heterogeneity in the included studies (I² values exceeding 50% in several analyses), which should be addressed more thoroughly.
- Some important demographic factors, such as socioeconomic status and racial/ethnic disparities, are not analyzed in depth.
- The exclusion of certain studies from the meta-analysis (e.g., those reporting on secondary lung cancer) should be explained with greater detail.
The discussion effectively interprets the results in relation to previous literature, but it could be strengthened in the following ways:
- Addressing the potential biases that might have influenced the findings (e.g., self-reported SHS exposure, publication bias).
- Exploring why some studies showed weaker or non-significant associations between SHS and lung cancer.
- Providing a more critical discussion on the limitations of measuring SHS exposure across different studies.
The conclusions summarize key findings and emphasize the importance of SHS reduction efforts. However, the recommendations remain broad. It would be beneficial to:
- Offer more specific policy recommendations based on the findings (e.g., stricter regulations on smoking in multiunit housing).
- Discuss the potential for future research, such as longitudinal studies or genetic predisposition to SHS-related lung cancer.
Suggested Revisions
- Clarify the research gap – Clearly define what distinguishes this meta-analysis from previous work.
- Justify database selection – Explain why the chosen databases were selected and whether any key sources might have been omitted.
- Improve heterogeneity analysis – Address the high I² values in several meta-analysis results and discuss their implications.
- Expand demographic analysis – Include more discussion on racial, socioeconomic, and gender differences in SHS exposure.
- Strengthen policy recommendations – Provide more concrete suggestions for public health interventions and legislative changes.
The study covers an important public health issue and follows rigorous systematic review and meta-analysis standards. However, significant revisions are necessary before publication. The article requires a clearer research gap, better justification for methodological choices, and a more nuanced discussion of results. If the authors implement these revisions, the study could make a meaningful contribution to tobacco control policies and lung cancer prevention.
Author Response
The study investigates the relationship between second-hand smoke (SHS) exposure and lung cancer risk among non-smokers in the United States through a systematic review and meta-analysis. The topic is highly relevant for public health and tobacco control policies. The authors follow PRISMA guidelines and conduct a meta-analysis using robust statistical methods. However, the paper has several areas that require improvement before it can be considered for publication.
The introduction presents a clear rationale for the study, emphasizing the health risks associated with SHS. The authors provide statistics on SHS exposure in the U.S. and outline its impact on different demographic groups. However, the introduction could be improved by:
Clearly defining the research gap (e.g., how this study builds upon previous meta-analyses).
Expanding on the novelty of the study in comparison to prior systematic reviews.
Providing a stronger justification for focusing solely on the U.S. rather than including global data.
We thank you for your valuable feedback. We appreciate the opportunity to improve the Introduction by clarifying the research gap, novelty, and study scope.
To address these concerns, we have updated the introduction section to define the research gap by clearly:
- Highlighting how previous meta-analyses on secondhand smoke (SHS) and lung cancer have not specifically analyzed exposure areas (e.g., childhood, household, workplace).
- Emphasizing that our study fills this gap by systematically examining which exposure settings contribute most to lung cancer risk.
- Comparing our study to prior reviews emphasizing the difference between them.
- Acknowledging that SHS exposure and smoking regulations vary globally and explaining that U.S. policies (e.g., workplace smoking bans, housing regulations) differ from those in other countries, making it necessary to limit the review to U.S.-specific data to be able to generate meaningful public health recommendations. Thus, including global data could introduce inconsistencies due to differences in smoking prevalence, regulatory frameworks, and cultural practices.
The methodology is thorough and follows PRISMA guidelines, which is a strength. However, there are several weaknesses:
The authors searched five databases, but there is no explanation of why these specific databases were chosen over others.
Thank you for your insightful comment. We acknowledge that our original manuscript did not explicitly justify the selection of the six databases used in our systematic review. To address this, we have revised the Methods section to provide a clear rationale for why these specific databases were chosen over others.
Our selection of databases ensured broad disciplinary coverage while maintaining a focus on high-quality, peer-reviewed research. We chose Google Scholar, PsycINFO, Scopus, PubMed, Web of Science, and IEEE Xplo based on each of their coverage of relevant research domains:
- PubMed – Covers biomedical and public health research, including epidemiological studies on SHS and lung cancer.
- Scopus & Web of Science – Index high-impact journals across multiple disciplines, ensuring inclusion of studies in epidemiology, occupational health, and environmental sciences.
- PsycINFO – Includes behavioral science and psychosocial studies, capturing research on SHS exposure behaviors, risk perception, and policy impacts.
- IEEE Xplore – Provides access to studies on environmental monitoring, air quality assessment, and occupational exposure, relevant to workplace SHS exposure.
- Google Scholar – Included to capture studies that may not be indexed in traditional databases, increasing search comprehensiveness.
The selection process for studies (including resolving reviewer disagreements) could be elaborated further.
Thank you for your valuable suggestion. We agree that the study selection process should be described in more detail for transparency. To address this, we have revised the Methods section to:
- Clarify the multi-stage selection process, including title-abstract screening, full-text review, and final inclusion.
- Explain how reviewer disagreements were handled, specifying that:
- Each study was independently reviewed by three authors.
- Discrepancies in study inclusion were resolved through discussion and, if needed, by consulting a fourth senior reviewer to reach consensus.
- Emphasize adherence to PRISMA guidelines, ensuring a rigorous and transparent selection process.
The use of statistical methods (random effects model, Hartung-Knapp method) is appropriate, but the rationale for choosing these specific methods should be better justified.
Thank you for your valuable feedback.We agree that a more explicit justification for the choice of statistical methods (random effects model and Hartung-Knapp method) is necessary. Please refer to the data analysis section to see the revised methodology with the details and justifications added.
The results section is well-structured and provides a clear summary of study identification, participant characteristics, and quality assessment. However, the following improvements are needed: There is a significant level of heterogeneity in the included studies (I² values exceeding 50% in several analyses), which should be addressed more thoroughly.
Thank you for your valuable feedback. We acknowledge that the high heterogeneity (I² > 50%) in several analyses should be addressed more thoroughly. To address that, we explicitly stated in the limitations some factors that could have contributed to the high heterogeneity in the articles (study design, measurement methods, etc.). We believe that the revisions provide a rigorous explanation of the high heterogeneity.
Some important demographic factors, such as socioeconomic status and racial/ethnic disparities, are not analyzed in depth.
Thank you for your feedback. While we believe in the importance of addressing demographic factors and disparities, it is noteworthy that not all included studies accounted for SES and racial factors consistently, limiting our ability to conduct a deeper statistical analysis across these variables. To specify that, we updated the results section to explicitly state the variability in how studies reported SES and racial data and acknowledge that due to inconsistencies in reporting, a pooled statistical analysis was not feasible.
The exclusion of certain studies from the meta-analysis (e.g., those reporting on secondary lung cancer) should be explained with greater detail.
Thank you for your insightful comment. We recognize the need to clarify the justification for excluding certain studies from the meta-analysis. We elaborated by explaining that studies focusing on secondary (metastatic) lung cancer were excluded because their outcomes do not represent the risk of developing lung cancer due to SHS exposure but rather the spread of cancer from other primary sites. It is noteworthy that the focus here is on the risk of developing SHS-induced primary lung cancer.
The discussion effectively interprets the results in relation to previous literature, but it could be strengthened in the following ways:
Addressing the potential biases that might have influenced the findings (e.g., self-reported SHS exposure, publication bias).
Exploring why some studies showed weaker or non-significant associations between SHS and lung cancer.
Providing a more critical discussion on the limitations of measuring SHS exposure across different studies.
Thank you for your feedback. We appreciate your suggestion to strengthen the discussion section by addressing potential biases, variations in the study findings, and limitations of SHS exposure measurement. To improve this section, we have discussed potential biases that could have influenced the results, including self-reported SHS exposure bias, publication bias, and selection bias in some study designs. We also explored the variability in study results, notably why some studies showed weaker or non-significant associations. We finally mentioned in the limitations that there was a difference in assessment methods and heterogeneity in exposure definitions across the studies.
The conclusions summarize key findings and emphasize the importance of SHS reduction efforts. However, the recommendations remain broad. It would be beneficial to:
Offer more specific policy recommendations based on the findings (e.g., stricter regulations on smoking in multiunit housing).
Discuss the potential for future research, such as longitudinal studies or genetic predisposition to SHS-related lung cancer.
Thank you for your valuable suggestions. To address your feedback, we have incorporated more specific policy recommendations and expanded the discussion on future research directions within the existing Practical Implications and Conclusions sections.
Suggested Revisions
Clarify the research gap – Clearly define what distinguishes this meta-analysis from previous work.
Justify database selection – Explain why the chosen databases were selected and whether any key sources might have been omitted.
Improve heterogeneity analysis – Address the high I² values in several meta-analysis results and discuss their implications.
Expand demographic analysis – Include more discussion on racial, socioeconomic, and gender differences in SHS exposure.
Strengthen policy recommendations – Provide more concrete suggestions for public health interventions and legislative changes.
The study covers an important public health issue and follows rigorous systematic review and meta-analysis standards. However, significant revisions are necessary before publication. The article requires a clearer research gap, better justification for methodological choices, and a more nuanced discussion of results. If the authors implement these revisions, the study could make a meaningful contribution to tobacco control policies and lung cancer prevention.
We thank you for your feedback. All the comments were addressed.
Reviewer 3 Report
Comments and Suggestions for Authors
In the concerned manuscript, the authors have summarized the outcome of secondhand smoking (SHS) in the outcome of lung cancer among non-smokers in US. This particular issue of SHS is a well-known for its effectivity in causing not only lung but other arrays of cancers. This manuscript, adhering to PRISMA guidelines, is an indicative of facts and figures of SHS and lung cancer in non-smokers.
However, there are certain areas that need to be clarified and/or addressed in a better way:
- There are some repetitions of references (for example, in the last paragraph of Introduction, line 59, 60 & 62, showed the same reference numbers).
- The authors could be clearer regarding the particularity of lung cancer and not allied smoking-induced cancers (like throat cancer).
- Authors could give some gross idea regarding the threshold exposure of SHS on which the inception of lung cancer is depending.
- A prior knowledge of 3-point system could be provided.
- The authors could give a detailed idea of pathological grades and types of lung cancer that are associated with SHS with respect to diverse arrays of studies.
Author Response
In the concerned manuscript, the authors have summarized the outcome of secondhand smoking (SHS) in the outcome of lung cancer among non-smokers in US. This particular issue of SHS is a well-known for its effectivity in causing not only lung but other arrays of cancers. This manuscript, adhering to PRISMA guidelines, is an indicative of facts and figures of SHS and lung cancer in non-smokers.
However, there are certain areas that need to be clarified and/or addressed in a better way:
- There are some repetitions of references (for example, in the last paragraph of Introduction, line 59, 60 & 62, showed the same reference numbers).
While we appreciate your feedback and have addressed the problem with repetitive references.
- The authors could be clearer regarding the particularity of lung cancer and not allied smoking-induced cancers (like throat cancer).
We appreciate the reviewer’s suggestion to clarify our focus on lung cancer and distinguish it from other smoking-induced cancers. To address this, we have revised the methods section to explicitly state that only studies investigating lung cancer were included in our review. We also updated the inclusion criteria to specify that studies focusing on other smoking-related cancers (e.g., throat, laryngeal, or esophageal cancers) were excluded. These clarifications ensure that our findings remain specific to the relationship between secondhand smoke (SHS) exposure and lung cancer risk.
- Authors could give some gross idea regarding the threshold exposure of SHS on which the inception of lung cancer is depending.
We appreciate the reviewer’s suggestion to discuss the threshold exposure of secondhand smoke (SHS) that may contribute to lung cancer risk. However, it is noteworthy that there is no universally agreed-upon threshold due to variations in exposure measurement methods across studies,
- A prior knowledge of 3-point system could be provided.
Please note that this is not required by the journal.
- The authors could give a detailed idea of pathological grades and types of lung cancer that are associated with SHS with respect to diverse arrays of studies.
We appreciate the reviewer’s suggestion to provide a detailed discussion of the pathological grades and types of lung cancer associated with secondhand smoke (SHS). However, our study primarily focuses on the overall association between SHS exposure and lung cancer risk rather than distinguishing between specific histological subtypes or pathological grades.
Many of the included studies did not report lung cancer subtypes consistently, and those that did often lacked uniform classification methods, making it challenging to draw robust conclusions across different pathological types. Additionally, our study aims to assess general risk estimates related to SHS exposure rather than delve into specific molecular or pathological mechanisms.
Round 2
Reviewer 2 Report
Comments and Suggestions for Authors
The authors have made the required corrections. I recommend the manuscript for publication.